# Lidar Depolarization Ratio of Atmospheric Pollen at Multiple Wavelengths

Stephanie Bohlmann<sup>1,2</sup>, Xiaoxia Shang<sup>1</sup>, Ville Vakkari<sup>3,4</sup>, Elina Giannakaki<sup>1,5</sup>, Ari Leskinen<sup>1,2</sup>, Kari Lehtinen<sup>1,2</sup>, Sanna Pätsi<sup>6</sup>, Mika Komppula<sup>1</sup>

<sup>1</sup> Finnish Meteorological Institute, Kuopio, 70211, Finland
 <sup>2</sup> Department of Applied Physics, University of Eastern Finland, Kuopio, Finland
 <sup>3</sup> Finnish Meteorological Institute, Helsinki, 00560, Finland
 <sup>4</sup> Atmospheric Chemistry Research Group, Chemical Resource Beneficiation, North-West University, Potchefstroom, South Africa

<sup>5</sup> Department of Environmental Physics and Meteorology, University of Athens, Athens, 15784, Greece <sup>6</sup> Biodiversity Unit, University of Turku, 20014 University of Turku, Finland

Correspondence to: Stephanie Bohlmann (stephanie.bohlmann@fmi.fi)

# Abstract.

Lidar observations during the pollen season 2019 at the European Aerosol Research Lidar Network (EARLINET) station in

- Kuopio, Finland were analyzed in order to optically characterize atmospheric pollen. Previous studies showed the detectability of non-spherical pollen using depolarization ratio measurements. We present lidar depolarization ratio measurements at three wavelengths of atmospheric pollen in ambient conditions. In addition to the depolarization ratio detected with the multiwavelength Raman polarization lidar Polly<sup>XT</sup> at 355 and 532 nm, depolarization measurements of a co-located HALO Photonics Streamline Doppler lidar at 1565 nm were utilized. During a four days period of high birch (*Betula*) and spruce
- (*Picea abies*) pollen concentrations, unusually high depolarization ratios were observed within the boundary layer. Detected layers were investigated regarding the share of spruce pollen to the total pollen number concentration. Daily mean particle depolarization ratios of the pollen layers on the day with the highest spruce pollen share are  $0.10 \pm 0.02$ ,  $0.38 \pm 0.23$  and  $0.29 \pm 0.10$  at 355, 532 and 1565 nm, respectively. Whereas on days with lower spruce pollen share, depolarization ratios are lower with less wavelength dependence. This spectral dependence of the depolarization ratios could be indicative of big, non-
- spherical spruce pollen. The depolarization ratio of pollen particles was investigated by applying a newly developed method and assuming a backscatter-related Ångström exponent of zero. Depolarization ratios of 0.44 and 0.16 at 532 and 355 nm for the birch and spruce pollen mixture were determined.

## **1** Introduction

Pollen is an essential part of plant reproduction as they store and transport genetic information of the plant. Because of the immobility of plants, pollen grains need to be transferred by other means for example wind or animals. Most grasses (*Poaceae*), conifers and around 10-20 % of flowering plants are wind-pollinated (Ackerman, 2000). Typical sizes of anemophilous pollen

grains range from 20-60  $\mu$ m (Shukla et al., 1998), but also large pollen grains like pine (*Pinus*) or spruce (*Picea*) pollen with diameters around 100  $\mu$ m are transported by wind. Because anemophilous plants produce pollen in very large numbers, they are almost omnipresent in the environment. Thus, it is not surprising that pollen have various effects on climate and human

health. Millions of people are suffering from pollen triggered allergies and the number is expected to rise due to the changing climate (Schmidt, 2016). Increasing temperatures and higher CO2 concentrations favor plant growth and pollen production. In regions with sufficient water resources, the growing seasons are therefore expected to become longer and the spatial distribution of allergenic species can be expected to change (Ziska and Beggs, 2012).

Pollen can be lifted up to several kilometers by turbulent mixing within the boundary layer. Airborne measurements revealed
the presence of pollen up to 4 km above ground with a considerable amount of pollen still observed at 2 km (Mandrioli et al., 1984; Rempe, 1937). Once reached the top of the boundary layer, pollen can be transported by wind over thousands of kilometers (Rousseau et al., 2008; Skjøth et al., 2007; Szczepanek et al., 2017). Dispersed in the atmosphere, they can affect the climate. By acting as ice nuclei pollen promote the formation of clouds (von Blohn et al., 2005; Diehl et al., 2001, 2002). Furthermore, pollen grains can rupture and lead to a high number of smaller particles. Those pollen fragments are efficient

cloud condensation nuclei (Steiner et al., 2015; Wozniak et al., 2018). An enhanced number of ice and cloud condensation nuclei can furthermore increase the cloud albedo and cloud lifetime and reduce precipitation.

Considering the relevance of pollen for climate and environment and the increasing importance as health hazard, it is of high importance to study the pollen distribution and transport mechanism in the atmosphere. To further improve pollen transport and forecast models, validation measurements are needed. Previously, we have shown that in the absence of other non-spherical

- particles, light detection and ranging (lidar) measurements and especially the particle depolarization ratio can be used to track pollen grains in the atmosphere (Bohlmann et al., 2019). We also estimated the depolarization ratio at 532 nm for pure birch (*Betula*) and pine pollen under certain assumptions as  $0.24 \pm 0.01$  and  $0.36 \pm 0.01$ , respectively (Shang et al., 2020). However, due to the measurement setup in our earlier studies, the lowest layers of the atmosphere (below 800 m), which naturally contain most of the airborne pollen, could not be investigated. In this study, we extend our dataset by operating an upgraded version
- of our multiwavelength Raman lidar Polly<sup>XT</sup> with four near-range channels at the same measurement site in Kuopio (Vehmasmäki), Finland. The improved lidar setup allowed measurements down to around 350 m above ground level. Furthermore, the new system measures the depolarization ratio also at 355 nm, while depolarization measurements were limited to 532 nm in the previous campaign. To further improve the investigation of pollen depolarization properties, we assessed the depolarization measurements of a co-located Halo Doppler lidar, which provides information at 1565 nm (Vakkari
- et al., 2020). Instrumentation and methods are presented in Section 2. The observation period is introduced in Section 3. Therein we present a case study and comparison of different pollen conditions during the period. Supporting measurement data, a statistical analysis and an estimation of pollen depolarization values are as well given in same section.

#### 2 Instrumentation and methods

Our measurement campaign was performed from 23 April to 28 August 2019 at the rural forest site in Vehmasmäki (Kuopio), Finland (62°44'N, 27°33'E, 190 m above sea level). Vehmasmäki (Kuopio) is part of the European Aerosol Research Lidar Network (EARLINET). The measurement site is permanently equipped with a Raman polarization lidar, a Doppler lidar, multiple in-situ instruments as well as standard meteorological instruments located on top of a measurement container. Furthermore, a 318 m tall mast provides measurements of temperature, relative humidity and wind at various levels from ground up to 300 m. During our measurement campaign, a Hirst-type spore trap was deployed on top of the container to measure the pollen concentration. In the following sub-sections, the instruments are introduced, and the layer detection method is explained.

## 2.1 Polly<sup>XT</sup> and processing of lidar observations

The multiwavelength Raman polarization lidar Polly<sup>XT</sup>-FMI (Engelmann et al., 2016) is a fully automated lidar instrument, which emits at three wavelengths, 355, 532 and 1064 nm, and is able to detect at two rotational-vibrational Raman channels,

- 387 and 607 nm and one water vapor detection channel at 407 nm in addition to the elastic backscatter channels at the emission wavelengths. The cross-polarized signal can be detected at 355 and 532 nm which allows the determination of the linear volume depolarization ratio (VDR) and linear particle depolarization ratio (PDR) at those two wavelengths. In addition to the far field channels, Polly<sup>XT</sup> includes a near-field unit with detection channels at 355, 387, 532 and 607 nm. The combination of near- and far-field channels allows the observation of continuous profiles down to about 120 m, the height of complete overlap
- of the near-field receiver. The initial vertical resolution of the acquired data is 7.5 m and the temporal resolution is 30 s. Further details on the instrument setup, principle and error propagation can be found in Engelmann et al. (2016). Near-realtime quicklooks are publicly accessible at the PollyNet website (<u>http://picasso.tropos.de/</u>, last access: 17/12/2020), which visualizes the measurements of all lidars belonging to PollyNet network (Baars et al., 2016).
- The lidar data was analyzed using the Klett-Fernald method (Fernald, 1984; Klett, 1981) whenever the weather conditions were adequate (i.e. no low clouds or rain) and the signal quality was sufficient for deriving high quality profiles. A constant lidar ratio of 60 sr was used to derive profiles of backscatter and extinction coefficients, based on measurements retrieved during night-time using the Raman method (Ansmann et al., 1990) and the results for birch-spruce mixtures of our previous study (Bohlmann et al., 2019). Far- and near-field profiles were merged to obtain the best possible profiles throughout the whole troposphere, enabling us to determine reliable profiles close to the ground. However, this is not the case for the 355 nm
- channel. Due to an overlap issue on the 355 nm far-range channel, reliable profiles at this wavelength were limited to 900 m above ground. The problem affected also the calculation of the depolarization ratio at 355 nm, which therefore could not be retrieved close to ground.

The profiles were temporally averaged in 2h intervals to match with the time resolution of the pollen samples. Aerosol layers within the lowest 4 km of the atmosphere were detected using an automatic layer detection algorithm using the gradient of the

- backscattered signal at 1064 nm. The local maximum of the first derivative is considered to be the layer bottom, while the height of the local minimum of the first derivative corresponds to the layer top (Belegante et al., 2014). Less smoothed profiles with a vertical smoothing range of 127 m were used to determine the correct geometrical layer boundaries without losing layer depth. Higher smoothing with a vertical range of 367 m, in turn, was applied to calculate reasonable optical properties within the layer. Intensive optical properties (lidar ratios, linear particle depolarization ratios, and Ångström exponents), commonly
- used for lidar-based aerosol characterization, were calculated for the automatically detected layers. Especially, the linear depolarization ratio and backscatter-related Ångström exponents (BÅE) are considered in this study. The linear depolarization ratio describes the change of the polarization state of the backscattered light. The strength of depolarization depends on the non-sphericity of the backscattering particles and the particle size relative to the wavelength (Sassen, 2005). The depolarization ratio increases with non-sphericity and can therefore be used to characterize the particle shape. The Ångström exponents
- describe the wavelength dependence of the scattering/extinction and relates to the particle size.

# 2.2 Halo Doppler lidar

In addition to the Polly<sup>XT</sup> lidar, a HALO Photonics Stream Line Pro Doppler lidar (Pearson et al., 2009) is operated permanently at our measurement site. This lidar operates at a wavelength of 1565 nm with a pulse repetition rate of 15 kHz and a pulse energy of 20  $\mu$ J. The minimum range of the instrument is 90 m and the range resolution is 30 m; other operating specifications

- can be found in Vakkari et al. (2020). This Halo lidar is equipped with a cross-polar receiver channel, which enables consecutive measurement of co- and cross-polar signals in vertically pointing mode and thus the retrieval of the depolarization ratio at 1565 nm. In this study, an integration time of 7 s was used, i.e. the co-polar signal is collected for 7 s followed by the cross-polar signal collected during the next 7 s. In addition to vertically pointing measurement, a conical scan lasting 2.5 min was scheduled every 15 min to retrieve the horizontal wind profile.
- The Halo Doppler lidar signal-to-noise ratio was post-processed according to Vakkari et al. (2019) and the depolarization ratio was processed similar to Vakkari et al. (2020). Bleed-through in the Halo internal polarizer was estimated from measurements at liquid cloud base as  $0.013 \pm 0.006$ , which is very close to the estimate of  $0.016 \pm 0.009$  from measurements in 2016 (Vakkari et al., 2020). The depolarization ratio was corrected for the bleed-through according to Vakkari et al. (2020).
- The retrieved depolarization ratio can be considered as linear particle depolarization ratio, since the contribution of molecular scattering at this long wavelength and low pulse energy is neglectable (Vakkari et al., 2020). Vertical wind speed from the copolar vertically pointing measurements and horizontal wind profile from the conical scans were used to calculate turbulent kinetic energy (TKE) dissipation rate following (O'Connor et al., 2010).

# 2.3 In-situ instruments

The pollen information was obtained using a Hirst-type volumetric air sampler located on top of the measurement container, 4 m above ground. The sampler collects particles onto an adhesive-coated tape which is manually analyzed afterwards. With

this type of sampler, it is possible to determine pollen types and concentrations with 2-hour time resolution. A more detailed description of the sampling principle can be found in Bohlmann et al. (2019) and Hirst (1952).

Various in-situ aerosol instruments were exploited for characterizing the overall aerosol load in the air at the site. An optical particle sizer (OPS; TSI Inc., model 3330) provided particle number concentration and size distribution in the size range of  $0.3-10 \mu m$ , which were used to calculate the volume size distribution (by assuming spherical shape for the particles). A

- Synchronized Hybrid Ambient Real-time Particulate Monitor (SHARP; Thermo Fisher Scientific Inc., model 5030) and an aethalometer (Magee Scientific, model AE-31) provided continuous total particle mass and black carbon (BC) concentration, respectively. The instruments resided in the container and sampled through vertical stainless-steel lines with total air inlets. Meteorological conditions, such as temperature, relative humidity, and wind speed and direction, may have effects on pollen
- concentration and distribution in the atmosphere. At Vehmasmäki these meteorological parameters are measured at several heights between 2 and 300 m on the co-located mast. The lowest height for wind speed and direction is 26 m, which is around the average canopy height at this site.

## **3** Results and discussion

During our four-month campaign in 2019, several periods of high pollen concentrations have been observed. One particular intense event with unusually high depolarization ratios and high pollen concentration will act as an example period for this study. An overview of this intense period is given in Fig. 1. From 16 – 19 May 2019, high bihourly concentrations of birch pollen, reaching up to 9000 grains m<sup>-3</sup>, were detected. The share of spruce pollen to the total pollen number concentration increased from 0.2% on 16 May to 21 % on 19 May. Other pollen types only accounted for 0.8% of the total pollen concentration during the period (Fig.1a) and thus can be neglected.

- The time-height display of Polly<sup>XT</sup> and Halo Doppler lidar observations is shown in Fig. 1 b-d and e, respectively. During daytime, a high aerosol load was observed within the first 2 km considering the strong range-corrected signal at 1064 nm (Fig. 1b). Also the VDR at 532 nm and the Halo particle depolarization ratio at 1565 nm show strongly enhanced values and are increasing towards the 19 May (Fig. 1d, e). At 355 nm, however, the VDR does not show significant features except a faint increase up to 2 km on 19 May. Faint lifted layers between 3 and 5 km were observed on all four days especially during night.
- Those layers seem to be weakly depolarizing. Backward-trajectories (not shown here) indicate that during all four days, the air masses arriving at different heights from 500 m to 5 km were advected from northwest and passed over North Scandinavia and the Norwegian Sea, i.e., they did not pass over dust influenced or highly polluted areas. Thus, it can be assumed that the studied air masses were not influenced by depolarizing dust. Elevated aerosol layers are likely lifted or long-range transported pollen.

#### 155 3.1 Highly depolarizing case study

The 19 May is here of special interest. Lidar measurements show a strongly increased backscatter coefficient at 1064 nm and increased volume depolarization ratios especially at 532 and 1565 nm (Fig. 1). The depolarization ratios at these wavelengths are exceptionally high compared to commonly observed depolarization ratios for non-spherical particles. However, measurements with the same measurement settings prior to the intense pollen period show usual values. We therefore consider our depolarization ratio measurements trustworthy.

- In Figure 2, the 19 May is presented separately, showing the range-corrected signal at 1064 nm (Fig. 2 a) and the volume depolarization ratio at 532 nm (Fig. 2 b). The period from 11:00 to 13:00 UTC (shown within white bars) was selected to present time averaged lidar profiles (Fig. 2 c-e). For processing the lidar measurements, a lidar ratio of 60 sr was applied as it was determined for birch and spruce mixtures in our previous study (Bohlmann et al., 2019). Relative humidity and temperature
- profiles according to GDAS1 (Global Data Assimilation System) data are shown in Fig. 2f. According to profiles of the backscatter signal and relative humidity, the top of the boundary layer is at around 2.5 km. Above 8 km cirrus clouds have been detected which show typical depolarization values for cirrus clouds of around 0.4 at 355 and 532 nm (Chen et al., 2002; Sassen and Zhu, 2009; Voudouri et al., 2020). Within the boundary layer the backscatter coefficient at 532 nm is larger than at 355 nm, resulting in a negative backscatter-related Ångström exponent (BÅE) at 355/532 nm of -0.64 ± 0.45. Cases of
- negative Ångström exponents have previously been observed for dust particles (Bohlmann et al., 2018; Rittmeister et al., 2017; Tsekeri et al., 2017; Veselovskii et al., 2016), and are likely caused by specific chemical compositions causing a spectral variation of the imaginary part of the refractive index (Veselovskii et al., 2016). The BÅE at 532/1064 nm, however, is positive,  $0.97 \pm 0.41$ , and comparable with values of dust smoke mixtures (Kanitz et al., 2014; Tesche et al., 2011). VDR and PDR values within the boundary layer are high. Mean PDRs are  $0.06 \pm 0.02$ ,  $0.46 \pm 0.22$  and  $0.35 \pm 0.07$  at 355, 532 and 1565 nm,
- respectively. While the depolarization ratios at 355 and 532 nm show a decline with height, the Halo PDR at 1565 nm is almost constant throughout the whole boundary layer. This may be caused by the higher sensitivity of the longer Halo wavelength to larger particles and the lower sensitivity to smaller particles, which would decrease the depolarization ratio. Most noticeable is the wavelength dependence of the depolarization ratio. A spectral dependence of the linear depolarization ratio of pollen has already been reported by Cao et al. (2010) who measured the depolarization ratio for different pollen types, such as birch and
- pine, in controlled laboratory experiments. For birch pollen Cao et al. (2010) reported a PDR of  $0.08 \pm 0.01$ ,  $0.33 \pm 0.00$  and  $0.30 \pm 0.01$  at 355, 532 and 1570 nm, respectively. No measurements have been conducted for spruce pollen. However, the authors state values for pine which belong to the same family as spruce and have smaller (60-74 µm) but in shape comparable pollen grains. According to Cao et al. (2010), depolarization ratios for pine pollen are  $0.20 \pm 0.01$ ,  $0.41 \pm 0.01$  and  $0.28 \pm 0.01$  at 355, 532 and 1570 nm, respectively. A similar wavelength dependence of the depolarization ratio with the peak at 532 nm
- was observed in our study for the spruce dominated aerosol mixture. The smaller depolarization at 355 nm and the higher depolarization at 532 and 1565 nm in our ambient measurements compared to the depolarization values for pure pine obtained in the laboratory, could be caused by the mixture of different pollen types in the atmosphere.

Very high depolarization ratios of aerosol particles have been detected with lidar before. Ansmann et al. (2009) measured depolarization ratios of 0.5-1 at 710 nm during the Saharan Mineral Dust Experiment (SAMUM). Those high depolarization ratios were caused by large dust, silt-sized and sand particles (> 10 µm) which were lifted by convective plumes and dust devils.

# 3.2 Case studies with different pollen concentrations

Figure 3 shows profiles of the Polly<sup>XT</sup> backscatter coefficients and particle depolarization ratio at 355 and 532 nm as well as the attenuated backscatter and PDR at 1565 nm by the Halo Doppler lidar for a selected 2-h time period on each day representing different pollen conditions. The bar chart illustrates the pollen concentration obtained from the Hirst-type pollen trap during the time period each day. Green color represents the birch pollen concentration, while red represents the spruce pollen concentration. On 19 May, the total pollen concentration and the share of spruce pollen were the highest. On 16 and 17 May, however, no spruce pollen was detected. A strong backscattering signal was detected up to 2.5 km on 19 May, whereas on the other days the signal was weaker, and according to the strong decrease of the backscatter signal, the boundary layer

height was lower. Most noticeable is the increase of the PDR at 532 and 1565 nm on the day with the highest spruce pollen share and total pollen concentration, while the PDR at 355 is around the same on all days and clearly lower than at the longer wavelengths.

The increased depolarization on 19 May is likely related to the large amount of highly non-spherical and large spruce pollen on this day compared to the other days. In Figure 4, micrographs of the pollen observations on 18 and 19 May from 9–11 UTC

- are shown. Birch and spruce pollen were observed on both days. Birch pollen grains are almost spherical, around 20-30 μm in diameter and have three pores on their surface. Spruce pollen grains, on the other hand, are non-spherical, and possess two air bladders which enable those large pollen grains (90-110 μm diameter on longest axis) to be dispersed by wind. The amount of both birch and spruce pollen was clearly higher on 19 May which may have resulted in the higher depolarization observed on this day. Certainly, other factors, such as ambient weather conditions or the fragmentation of pollen grains, could have
- contributed to this observation as well. Those effects will be investigated in the following section.

## 3.3 Findings from supporting data

Pollen release and distribution is heavily influenced by the ambient weather conditions. Higher temperature and wind speed positively affect the pollen concentration while the relative humidity and pollen concentration are negative related (Bartková-Ščevková, 2003; Noh et al., 2013). Under certain meteorological conditions, e.g. humidity changes and turbulent disturbances,

pollen grains can rupture to smaller fragments (Hughes et al., 2020; Miguel et al., 2006; Taylor et al., 2002, 2004). The pollen fragments are not recorded as pollen, and therefore do not contribute to the pollen count using microscopic analysis. However, the fragments can persist much longer in the atmosphere than intact pollen due to their smaller size and lower falling velocity. The small pollen fragments can act efficiently as ice nuclei (von Blohn et al., 2005; Diehl et al., 2001, 2002) and penetrate deeper into the respiratory system than intact pollen where they can trigger severe allergenic reactions and asthma.

- Figure 5 gives an overview of the meteorological conditions during the studied period. The temperature and relative humidity at 2 m (Fig. 5b) and wind speed, wind direction and gust wind at 26 m (Fig. 5c) above ground are shown in addition to the pollen concentration (Fig. 5a). The diurnal cycle of temperature and relative humidity are clearly visible with a maximal relative humidity during the night and maximal temperature shortly after noon. Wind measurements show only a weak diurnal change with a slightly lower wind speed in the evening hours. On 19 May, the wind speed increased rapidly before noon and
- a more distinct diurnal cycle compared to the other days is apparent. At the same time, the temperature was higher and the relative humidity slightly lower than on the preceding days, which could explain the higher pollen concentration on this day. The local wind direction changed from northwest on 16 May to southeast on 19 May. Gust winds predominately follow the trend of the mean wind speed, although on 17 May, the stronger gust winds at around 19 UTC could explain the high amount of birch pollen at this time.
- Turbulent movements within the boundary layer can affect the dispersal of airborne pollen, and pollen grains can be lifted vertically to heights where they are detectable with lidar. The turbulent kinetic energy (TKE) dissipation rate obtained from the Halo Doppler lidar is shown in Fig. 5d, but no significant difference between the four days was found to further explain the higher depolarization ratio. The higher detection height on 19 May is caused by a higher aerosol load and therefore higher signal-to-noise ratio on that day.
- In-situ measurements have been checked to characterize the ground-level aerosol conditions at our measurement site. The particle number concentration at ground level was higher on 18 and 19 May than on 16 and 17 May (Fig. 6a). The evolution of the volume size distribution (Fig. 6b) shows an increase in the concentration of 3-10  $\mu$ m particles, which could be fragments of birch or spruce pollen. The increase in the coarse particle fraction was verified with the total mass concentration (data not shown). The BC concentration (not shown) was low (around 0.1-0.2  $\mu$ g/m<sup>3</sup>) throughout the period, except for a short peak on
- 18 May at 18 UTC (also visible in the OPS data, Fig. 6a), thus no smoke contribution in the boundary layer is expected.

## 3.4 Lidar derived optical parameters

To investigate the relationships between lidar-derived optical parameters and the pollen observations and to define characteristic values for specific pollen conditions, we determined mean values in the detected layers shown in Fig. 1f. We here focus on the lowest detected layer as it can be assumed to contain most of the local pollen concentration. A total of 47

- pollen layers has been detected in the 2-hour averaged dataset of the 4-day period. Mean values of the lidar derived optical properties are presented in Figs. 7 and 8. Color represents the share of spruce pollen of the total pollen number concentration detected by the Hirst-type spore trap. Size of the markers represents the total 2-hour pollen concentration. Daily mean values are shown with black markers, error bars represent the standard deviation. The lower limit for reliable profiles was about 350 m after combining near- and far-field channels and vertical smoothing; whereas a higher layer bottom limit of 900 m was
- applied on the profiles of PDR at 355 nm (Fig. 7a) and the Ångström exponent at 532/1064 nm (Fig. 8b), due to the overlap problems at 355 nm and the higher full overlap height at 1064 nm, respectively.

In Figure 7, the particle depolarization ratios at all three wavelengths are compared. Two groups depending on the spruce pollen share are apparent in the comparison of PDR at 532 against 355 nm (Fig. 7a) and PDR at 532 against 1565 nm (Fig. 7b). Cases with a high share of spruce pollen and high total pollen concentration tend to show the highest PDR values ranging

- between 0.4 0.9 at 532 nm, while the PDR for cases with a lower share of spruce pollen is below 0.4. The PDR at 355 nm is below 0.17 for all cases, and no correlation with the spruce pollen share is apparent. The relationship between the PDR at 1565 and 532 nm is nearly linear when the spruce share is low (Fig. 7b). With increasing spruce pollen share, the PDR at 532 nm exceeds the PDR at 1565 nm. This wavelength dependency of the PDR with increasing spruce pollen share could be characteristic for this large and non-spherical pollen type and provide a means for classifying conifer pollen.
- Figure 8 shows the backscatter-related Ångström exponents at 355/532 nm and 532/1064 nm against the PDR at 532 nm. The tendency towards smaller Ångström exponents with increasing depolarization ratio indicates the increasing impact of larger and non-spherical particles such as spruce pollen. Similar behavior has already been reported in Bohlmann et al. (2019) and Shang et al. (2020). For cases with a high amount of large non-spherical spruce pollen, e.g., with >15% share of the total pollen number concentration, the BÅE at 355/532 nm is negative. Previously, negative Ångström exponents have only been reported
- for dust close to the dust source region where very large dust particles are present (Hofer et al., 2020; Rittmeister et al., 2017; Veselovskii et al., 2016). This feature, however, could also be characteristic for aerosol conditions with the presence of very large pollen grains.

Table 1 summarizes the daily mean values shown in Figure 7 and 8 with black markers. Mean values are given for different layer bottom heights as profiles could only be retrieved down to 900 m at 355 and 1064 nm. Noteworthy is the spectral

- dependence of the depolarization ratio, which is also illustrated in Fig. 9. Especially on 19 May a higher depolarization ratio was observed at 532 than at 1565 nm, whereas the depolarization ratios at 532 and 1565 nm are almost equal on the preceding days. The PDR at 355 nm is around 0.1 on all days. The number of observed spruce pollen increased each day. Longer wavelengths are more sensitive to bigger particles and less influenced by Rayleigh scattering (Ansmann et al., 2009). The strong spectral dependency on 19 May can therefore be explained by the high amount of non-spherical spruce pollen on that
- 275 day. For comparison, laboratory-derived depolarization ratios by Cao et al. (2010) are shown in Fig. 9. Empty markers represent the lidar depolarization ratios for paper birch (*Betula papyrifera*) and Virginia pine (*Pinus virginiana*) measured in a controlled laboratory environment. Especially, the spectral dependence for pine pollen is very similar to our observations on the spruce dominated day. It must be kept in mind that the laboratory measurements were performed in an aerosol chamber and that they represent the depolarization ratio of the pure and dry pollen. Furthermore, pollen from different species than we
- detected at our measurement site were used. Different values were therefore expected. However, the incidence that we detected similar wavelength dependence in ambient conditions as in the laboratory measurements, suggests that the wavelength dependence with maxima at 532 nm might be characteristic for pollen and especially for large and non-spherical conifer pollen types.

Also, the daily mean backscatter-related Ångström exponent is clearly different on the spruce dominated day. While it is around 0.9-1.3 at 355/532 nm and 1.18-1.52 at 532/1064 nm on the days with low spruce pollen share, BÅEs are clearly smaller

at both wavelength pairs on the spruce dominated day, suggesting the presence of larger particles. The negative BÅE at 355/532 nm might be characteristic for large, non-spherical spruce pollen grains since it has only been observed on days with a high spruce pollen concentration. It must be noted, that the Ångström exponent also depends on the background aerosols and its use to characterize pollen needs to be considered carefully. However, under the same background conditions, the Ångström exponent can be an indicator for the pollen type. As the considered time period is rather short and no far-range transported aerosol sources were observed, the background conditions can be assumed stable during the period. The change of the Ångström exponent can therefore be attributed to the presence of pollen.

#### 3.5 Estimation of pollen depolarization values

To estimate the depolarization values for pure pollen, a newly developed method by Shang et al. (2020) was applied to the lidar measurements. Mathematically, there is a power law relationship between the backscatter-related Ångström exponent of total particles and the pollen backscatter contribution (the ratio of pollen backscatter coefficient and the total particle backscatter coefficient). The depolarization ratio at 532 nm of pure pollen was investigated using the BÅE at 355/532 nm. Our results on this intense pollen event show two groups of data depending on the spruce pollen share: i) a quite good power law relationship is found for cases with a low share of spruce pollen; ii) the BÅE at 355/532 nm is almost constant with increasing

- PDR for cases with higher share of spruce pollen. These two groups can also be identified as the two modes in the frequency distribution of BÅE values at 355/532 nm presented in Fig. 10. The first mode of values around -0.9, denoted as "extreme cases", are related to PDR bigger than 0.25, mostly for the data on the last day (high spruce share). For these cases the BÅE is almost constant throughout the boundary layer whereas the depolarization ratio is decreasing with height. This reveals the limitation of Ångström exponent at lidar wavelengths of 355 and 532 nm for the detection of big pollen particles since the 305 BÅE normally decreases with increasing share of pollen particles, i.e. increasing PDR.
- A threshold of -0.4 was applied to remove those "extreme cases" before applying the algorithm of Shang et al. (2020). A depolarization ratio of 0.03 at both 355 and 532 nm was used for non-pollen particles, which is a mean value for pollen-free periods at our measurement site. After the iteration, we found the relationship between the pollen BÅE at 355/532 nm and the pollen depolarization ratio at 532 nm presented in Fig. 11. This relationship is representing the mixture of birch and spruce
- pollen, with low spruce pollen share after the selection of the threshold. Our previous studies (Bohlmann et al., 2019; Shang et al., 2020) indicate that pollen are medium to high absorbing particles with lidar ratio values from 55 to 70 sr at both 355 and 532 nm for birch and spruce pollen, without significant wavelength-dependence. The pollen grains are quite big, and thus can be assumed to be wavelength independent on the backscatter at wavelengths of 355 and 532 nm. Under the assumption that the backscatter-related Ångström exponent at 355/532 nm should be zero for pollen, the pollen depolarization ratio was found
- to be 0.44 (Fig. 8), for the mixture of birch and spruce pollen observed during this intensive period of high pollen concentrations. The depolarization at 355 nm was calculated using the pollen backscatter contribution at 355 nm and the BÅE at 355/532 nm. Assuming a BÅE of zero, a pollen depolarization ratio of 0.16 at 355 nm was found for the birch and spruce mixture. In order to retrieve the pollen depolarization ratio of the "extreme cases", we used the BÅE at 532/1064 nm as longer