# Peer review of "Lidar Depolarization Ratio of Atmospheric Pollen at Multiple Wavelengths"

_Atmospheric Chemistry and Physics, 2020_

## Referee Comment (RC1) · Anonymous Referee #1 · 12 Jan 2021

Title: Lidar Depolarization Ratio of Atmospheric Pollen at Multiple Wavelengths Paper No.: MS No.: acp-2020-1281

Revision of the paper

Anonymous Referee

Comments: This paper is a study on the change of the depolarization ratio and the backscatter Angstrom exponent measured by the multi-wavelength Raman lidar the Doppler lidar as spruce pollen increases when birch pollen is the main case. In addition to the 532 nm wavelength, which is mainly used to measure the polarization extinction of pollen, it is judged to be an unusual study that simultaneously measured the polarization extinction at 355 nm and 1565 nm. As mentioned in the conclusion

of the paper, more research is required in the future to provide information on pollen types and concentrations through LiDAR observation, but the current research results in this paper are expected to be used as important data for the next pollen research. This paper is considered appropriate for publication on the ACP. However, it is judged that there are some check points. A detailed review follows:

Minor Comments 1. Page 4. Line 106, 110, and 115: Author mentioned Dopper lidar in the paper. But, expressions in line 106, 110, and 115 are different. Please check it. 2. Line 153: "Elevated aerosol layers are likely lifted or long-range transported pollen.". Could you add in additional explanations to explain the content of this sentence?

Suggestion 1. Rupture of pollen grains is important in this paper. Can you add a picture of the fragment pollen ruptured by weather conditions etc? 2. How about the change the scale of PDR 355 in Figure 3 (b)?

---

## Referee Comment (RC2) · Anonymous Referee #2 · 26 Jan 2021

Lidar observations of pollen in the atmosphere receive interest by the community as pollen are not yet as well characterized as other aerosol types. The authors contribute to the characterization effort by extending the wavelength range from 355 and 532 nm towards 1565 nm and by presenting measurements of birch-spruce pollen mixtures. Especially, the spruce pollen turns out to be strongly depolarizing. Especially the spectral dependence of the depolarization ratio is of great value. The figures of the manuscript are in an excellent shape. However, I have two major concerns which should be addressed before publication. In the following, I'll provide my comments to improve the manuscript.

My major concerns:

1. Presenting a particle linear depolarization ratio (PLDR) of a birch-spruce pollen

mixture (44% at 532 nm and 16% at 355 nm) has not so much use to the community. We need to have the same mixture of birch and spruce pollen to measure the same PLDR. The PLDR of pure birch and pure spruce pollen is needed to apply the findings to future measurements.

Under the assumption that the mixing ratio measured by the in-situ pollen sampler is valid for the whole boundary layer, the PLDR of each pure type could be calculated. This assumption is already made throughout the paper (line 143, Fig. 7+8), but contains some uncertainties which should be addressed.

Furthermore, Shang et al. (2020) already reported PLDR for pure birch pollen. This open another way to address the PLDR of pure spruce pollen. Reporting a value for the PLDR of pure spruce pollen will enhance the impact of this manuscript.

2. The calibration of the PLDR is not touched in the manuscript sufficiently. This is an essential part for a paper focusing on PLDR values. Otherwise the PLDR values cannot be trusted. Two lidars are involved which have to be calibrated. The short period of time (4 days) allows to give more detail about the calibration measurements of the PollyXT and the Halo Doppler lidar during this specific period.

Using the depolarization ratio measurements of a Halo Doppler lidar is something new. The paper by Vakkari et al. (2020) describing the depolarization calibration is still under review and needs to be accepted before acceptance of the present manuscript is possible. Nevertheless, some more details about the depolarization calibration of the Halo Doppler lidar during the pollen campaign are necessary for this paper.

The PLDR at 532 nm decreases with height for all days whereas the depolarization ratio at 1565 nm stays vertically almost constant (Fig. 3). Why? This is not satisfactorily explained (line 176/177) and could lead to the impression that something is wrong with the depolarization ratio of the Doppler lidar.

My main comments:

3. A subsection describing the birch and spruce pollen is missing. Fig. 4 and the describing text in Sect. 3.2 occur somewhere in the description of the lidar measurements and should be moved to the beginning of the paper.

4. Do you mainly observe whole pollen grains or fragments?

A spruce pollen with 100 $\mu$m in size is large compared to all wavelengths used in your study and so no spectral behavior is expected. The strong difference in PLDR between 355 and 532 nm could not be explained with such large pollen grains. Your optical particle sizer (OPS) counts only up to 10 $\mu$m, which is even too small for birch pollen (20 − 30 $\mu$m in diameter). Do you have any indication about the strength of the fragmentation process? Is it related to temperature, RH, wind speed or age of the pollen grain?

5. Do the depolarization ratios reported by Cao et al. (2010) correspond to whole pollen grains or fragments?

6. Does the microscopic analysis of the Hirst sampler count only whole pollen grains or does it include fragments as well? Maybe the large spruce pollen grains are more likely fragmented. This could explain their high PLDR as you discuss in lines 330-333. Please extend the discussion and collect further information on fragmented pollen grains. Maybe on the tape of the Hirst sampler?

7. Please improve your discussion of the spectral dependence of the PLDR and motivate, why it is important to measure the depolarization ratio at several wavelengths. The spectral dependence of the PLDR was studied previously for different aerosol types. A similar behavior with a maximum at 532 nm and a decrease towards the ultraviolet and the near infrared was observed for mineral dust (Burton et al., 2015, Haarig et al., 2017). The optical properties of mineral dust are dominated by the large particles (>1 $\mu$m) as well. Whereas small smoke particles in the stratosphere (< 1$\mu$m) show a completely different behavior with a strong decrease of the PLDR with increasing wavelength (Haarig et al., 2018, Hu et al., 2019).

8. The depolarization ratio at 1565 nm is consequently not called particle depolarization ratio (PDR), but just depolarization ratio (DR). This might confuse some readers. In fact, you measure the volume depolarization ratio at 1565 nm. However, the molecular influence is negligible at such long wavelengths and the volume depolarization ratio is equal to the particle depolarization ratio. Can you calculate once the molecular influence at 1565 nm? Are gas absorption lines expect at this wavelength which could interfere with your results? With this low number you can argue, that you measure the PDR.

9. You are arguing that the pollen grains are large compared to the wavelength (355 and 532 nm), so the BAE = 0. But how do you explain the large difference in the PLDR between 355 and 532 nm?

Minor comments

10. Ice nuclei (e.g., L.43) should be called "ice nucleating particles" (INP) to follow the convention of Vali et al., 2015. Please consider this throughout the manuscript.

11. L.39/40 "Airborne measurements revealed the presence of pollen up to 4 km above ground with a considerable amount of pollen still observed at 2 km" Where did they measure? Please include the region at this point.

12. L.45 The phrase "increase the cloud albedo and cloud lifetime and reduce precipitation" goes beyond the scope of your article and would need further references.

13. L.88 Did you merge the signals or the products? Please be more specific.

14. L.116 "was processed similar to Vakkari et al. (2020)" Where is the difference to Vakkari or is it just the different bleed-through? When did you perform the calibration at liquid cloud base?

15. L.148 The VDR in the UV is strongly influenced by the molecular scattering and therefore the effect of the particles is smaller compared to larger wavelengths.

16. L.155 Find a better title for the subsection. Something like "Case study with highly depolarizing aerosol".

17. Section 2.1, 3.1 and 3.2 needs some more structure, e.g., Section 2.1 starts and ends with depolarization. Better separate the discussion about the negative BAE from the spectral dependence of PLDR in Section 3.1. In Section 3.2 move the pollen description to a separate (sub)section at an earlier position in the paper.

18. Which trees are surrounding your measurement site? Do they contribute to the pollen load?

19. L.288 "It must be noted, that the Ångström exponent also depends on the background aerosols and its use to characterize pollen needs to be considered carefully". It is mentioned very late in the manuscript. At the beginning you should make clear, that BAE and PLDR are always a mixture of pollen and background aerosol. Already there, you can mention that you will later use the method of Shang et al. (2020) to separate the background and the pollen influence.

20. Fig. 3(e) Consider adjusting the x-axis to a maximum value of 2 to be comparable to the other backscatter coefficient profiles and enlarge the profiles for 16 – 18 May.

21. Fig. 4 Add the year to the dates in the caption.

22. Fig. 10 + 11 Please add the threshold (-0.4) to the figures.

23. The text should be improved in terms of spelling and unusual expressions, e.g.,

L.242 relationship, no plural.

L.293 "Estimation of the pollen depolarization ratio"

L.296 "of total particles" – find a better expression.

L.302 "are related to PDR larger than"

L.329 has –> have

L.350 affect the detected depolarization ratio as well.

24. Avoid long sentences, better split them, e.g., L.85-88; 93-95 (using . . . using)

References

Burton, S. P., Hair, J. W., Kahnert, M., Ferrare, R. A., Hostetler, C. A., Cook, A. L., Harper, D. B., Berkoff, T. A., Seaman, S. T., Collins, J. E., Fenn, M. A., and Rogers, R. R.: Observations of the spectral dependence of linear particle depolarization ratio of aerosols using NASA Langley airborne High Spectral Resolution Lidar, Atmos. Chem. Phys., 15, 13453–13473, https://doi.org/10.5194/acp-15-13453-2015, 2015.

Cao, X., Roy, G., and Bernier, R.: Lidar polarization discrimination of bioaerosols, Opt. Eng., 49, 116201, https://doi.org/10.1117/1.3505877, 2010. 

Haarig, M., Ansmann, A., Althausen, D., Klepel, A., Groß, S., Freudenthaler, V., Toledano, C., Mamouri, R.-E., Farrell, D. A., Prescod, D. A., Marinou, E., Burton, S. P., Gasteiger, J., Engelmann, R., and Baars, H.: Triple-wavelength depolarization-ratio profiling of Saharan dust over Barbados during SALTRACE in 2013 and 2014, Atmos. Chem. Phys., 17, 10767–10794, https://doi.org/10.5194/acp-17-10767-2017, 2017.

Haarig, M., Ansmann, A., Baars, H., Jimenez, C., Veselovskii, I., Engelmann, R., and Althausen, D.: Depolarization and lidar ratios at 355, 532, and 1064 nm and microphysical properties of aged tropospheric and stratospheric Canadian wildfire smoke, Atmos. Chem. Phys., 18, 11847–11861, https://doi.org/10.5194/acp-18-11847-2018, 2018.

Hu, Q., Goloub, P., Veselovskii, I., Bravo-Aranda, J.-A., Popovici, I. E., Podvin, T., Haeffelin, M., Lopatin, A., Dubovik, O., Pietras, C., Huang, X., Torres, B., and Chen, C.: Long-range-transported Canadian smoke plumes in the lower stratosphere over northern France, Atmos. Chem. Phys., 19, 1173–1193, https://doi.org/10.5194/acp-19-1173-2019, 2019.

Shang, X., Giannakaki, E., Bohlmann, S., Filioglou, M., Saarto, A., Ruuskanen, A.,

Leskinen, A., Romakkaniemi, S., and Komppula, M.: Optical characterization of pure pollen types using a multi-wavelength Raman polarization lidar, Atmos. Chem. Phys., 20, 15323–15339, https://doi.org/10.5194/acp-20-15323-2020, 2020.

Vakkari, V., Baars, H., Bohlmann, S., Bühl, J., Komppula, M., Mamouri, R.-E., and O'Connor, E. J.: Aerosol particle depolarization ratio at 1565 nm measured with a Halo Doppler lidar, Atmos. Chem. Phys. Discuss. [preprint], https://doi.org/10.5194/acp-2020-906, in review, 2020.

Vali, G., DeMott, P. J., Möhler, O., and Whale, T. F.: Technical Note: A proposal for ice nucleation terminology, Atmos. Chem. Phys., 15, 10263–10270, https://doi.org/10.5194/acp-15-10263-2015, 2015.

---

## Author Response (AR1)

**Letter of Reply to Referee 1**

Thank you for carefully reading the manuscript and providing useful suggestions to improve the paper.

The replies to your comments are given below. Changes in the manuscript are marked in the revised manuscript.

**Minor Comments**

**1. Page 4. Line 106, 110, and 115: Author mentioned Doppler lidar in the paper. But, expressions in line 106, 110, and 115 are different. Please check it.**

Thank you for that comment. It has been changed to the full name "Halo Photonics Stream Line Doppler lidar" or "Halo Doppler lidar" throughout the manuscript.

**2. Line 153: "Elevated aerosol layers are likely lifted, or long-range transported, pollen.". Could you add in additional explanations to explain the content of this sentence?**

The layers that we observed between 3-5km are probably pollen which have been lifted by turbulence or long-range transported pollen as other sources of depolarizing aerosols are missing along the trajectories which only passed over Northern Europe and the Norwegian Sea.

**Suggestions**

**1. Rupture of pollen grains is important in this paper. Can you add a picture of the fragment pollen ruptured by weather conditions etc?**

Pollen rupture is indeed important when investigating the depolarization caused by pollen in the atmosphere. However, we don't have the necessary information about the amount of pollen fragments to characterize the impact of pollen fragments on our results (please also see comment 4 and 6 of Referee 2). We decided not to add a picture of pollen fragments in order not to emphasise the ruptured pollen in this paper. The impact of pollen fragments on lidar measurements could be an objective for future studies.

**2. How about the change the scale of PDR 355 in Figure 3 (b)?**

It is true that changing the scale of Fig.3b would increase the visibility of the profiles, however we would like to keep the scale subplots axis consistent. In this way the profiles at the different wavelengths are better comparable and the difference is clearly visible.

**Letter of Reply to Referee 2**

Thank you for carefully reading the manuscript and providing useful suggestions to improve the paper.

Replies to your comments are given below and the changes are marked in the revised manuscript.

**Major concerns:**

**1. Presenting a particle linear depolarization ratio (PLDR) of a birch-spruce pollen mixture (44% at 532 nm and 16% at 355 nm) has not so much use to the community. We need to have the same mixture of birch and spruce pollen to measure the same PLDR. The PLDR of pure birch and pure spruce pollen is needed to apply the findings to future measurements. Under the assumption that the mixing ratio measured by the in-situ pollen sampler is valid for the whole boundary layer, the PLDR of each pure type could be calculated. This assumption is already made throughout the paper (line 143, Fig. 7+8), but contains some uncertainties which should be addressed. Furthermore, Shang et al. (2020) already reported PLDR for pure birch pollen. This open another way to address the PLDR of pure spruce pollen. Reporting a value for the PLDR of pure spruce pollen will enhance the impact of this manuscript.**

We agree that depolarization ratios of pure spruce and birch pollen would be very valuable for the community and deriving those values is the final goal of our pollen studies. We are working on this and are collecting more data to be able to reveal the properties for different pure pollen types. However, there are difficulties and uncertainties which prevent us to provide conclusive values at this point using current datasets. One is the assumption of a constant mixing ratio of background aerosol and different pollen types (e.g. birch and spruce) throughout the boundary layer. The estimated values for birch reported by Shang et al. (2020), is based on a pollen period when almost only birch is present in the atmosphere. Until now, we haven't found a good "spruce pollen period" to study the spruce pollen alone. During our previous campaigns in Finland, the spruce pollen is often presented together with birch pollen. The mixing of spruce and birch pollen in the air makes the retrieval of pure pollen properties much more challenging. Nevertheless, we believe that the optical properties of this birch-spruce pollen mixture and the evidence of high depolarization ratios caused by spruce pollen can provide a good database in the literature and will be valuable for future pollen studies. Furthermore, this is the first time depolarization ratios at 3 wavelengths are reported for a pollen mixture, the spectral dependence showing in this paper is of high interest for the community (we have emphasised this part in section 3.5 of the revised manuscript).

We have performed some analysis following the referee's suggestion. 2 steps are needed to estimate PLDR of pure pollen:

1. Separation of pollen mixture and background aerosols, we need to assume a PLDR of the pollen mixture which is unknown.
2. Separation of birch and spruce pollen from the pollen mixture profile. This step is easy as we know the PLDR of birch pollen.

The problem is mostly in the step 1, because the background and pollen mixture contributions are unknown. The retrieval of pure pollen properties in this case is a separation of 3 aerosol types, which is much more complicated than the one given in Shang et al. (2020).

We assumed that the mixing ratio of the volume concentration of pollen (assuming spherical shape) measured by the in-situ pollen sampler is constant for the whole boundary layer, and we assumed a value for PLDR of the pollen mixture, we retrieve very high depolarization ratios which, in our opinion, need further validation before being published with confidence. We hope that we will observe pure spruce cases without contribution of other pollen types in future studies, so that the real influence of pure spruce pollen can be determined.

**2. The calibration of the PLDR is not touched in the manuscript sufficiently. This is an essential part for a paper focusing on PLDR values. Otherwise the PLDR values cannot be trusted. Two lidars are involved which have to be calibrated. The short period of time (4 days) allows to give more detail about the calibration measurements of the PollyXT and the Halo Doppler lidar during this specific period. Using the depolarization ratio measurements of a Halo Doppler lidar is something new. The paper by Vakkari et al. (2020) describing the depolarization calibration is still under review and needs to be accepted before acceptance of the present manuscript is possible. Nevertheless, some more details about the depolarization calibration of the Halo Doppler lidar during the pollen campaign are necessary for this paper. The PLDR at 532 nm decreases with height for all days whereas the depolarization ratio at 1565 nm stays vertically almost constant (Fig. 3). Why? This is not satisfactorily explained (line 176/177) and could lead to the impression that something is wrong with the depolarization ratio of the Doppler lidar.**

The Polly$^{XT}$ depolarization measurement are calibrated using the Δ90° method (Freudenthaler, 2016). More details about the Polly$^{XT}$ depolarization calibration were added in Section 2.1 in lines 97-102:

*"To obtain high quality depolarization measurements, calibration measurements are automatically performed three times a day using the Δ90° method (Freudenthaler, 2016; Freudenthaler et al., 2009). A calibration constant $V_\lambda$\* determined from two subsequent measurements with a difference of exactly 90° is used to calibrate the volume depolarization ratio. During the presented time period, quite stable calibration constants V\* of 1.85\*10-1 ± 4.01\*10-3 at 355 nm and 6.90\*10-3 ± 6.29\*10-5 at 532 nm were determined and applied to calibrate depolarization measurements."*

More details about the Halo depolarization calibration using measurements at liquid cloud base are now given in Section 2.2 in lines 125-133:

*"The Halo Doppler lidar signal-to-noise ratio was post-processed according to Vakkari et al. (2019) and the depolarization ratio was processed according to Vakkari et al. (2020). The design of Halo Doppler lidars does not include a depolarization calibration unit nor does it allow to add one. Therefore, depolarization measurements can only be calibrated using the depolarization ratio at liquid cloud base. As liquid cloud droplets do not polarize the backscattered light, non-zero depolarization ratios at cloud base indicate incomplete extinction, called bleed-through, in the internal polarizer (Vakkari et al., 2020). The bleed-through in the Halo internal polarizer was estimated from liquid cloud base observed at 1200-1400 m a.g.l. on 14 June 0:30-3:30 UTC as 0.013 ± 0.006. This estimate is very close to the estimate of 0.016 ± 0.009 from measurements with same instrument in 2016 (Vakkari et al., 2020). The depolarization ratio was corrected for the bleed-through according to Vakkari et al. (2020)."*

We note that Vakkari et al. (2020) has now been accepted for publication and based on the review no modifications are needed for the Halo calibration here.

The explanation of the almost constant depolarization ratio at 1565 nm has been improved and is given in line 208-213:

*"In our observations, the depolarization ratios at 355 and 532 nm show a decline with height, while the Halo PDR at 1565 nm is almost constant throughout the whole boundary layer. Values of PDR, as well as BÅE, always represent a mixture of pollen and background aerosol. We assume that the concentration of pollen decreases with height, i.e. increasing distance to the pollen source, and the influence of background aerosol increases. The longer Halo wavelength is more sensitive to larger particles (pollen) and less sensitive to smaller particles (background aerosol) and thus the depolarization ratio does not decrease with height in contrast to the PDR at 355 and 532 nm."*

**Main comments:**

**3. A subsection describing the birch and spruce pollen is missing. Fig. 4 and the describing text in Sect. 3.2 occur somewhere in the description of the lidar measurements and should be moved to the beginning of the paper.**

A subsection describing the pollen was added as section "2.3 Pollen observations" as follows:

*"The pollen information was obtained using a Hirst-type volumetric air sampler located on top of the measurement container, 4 m above ground. The sampler collects particles down to a size of 3.7 μm (Muilenberg, 2003) onto an adhesive-coated tape which is manually analyzed afterwards. With this type of sampler, it is possible to determine pollen types and concentrations with 2-hour time resolution. A more detailed description of the sampling principle can be found in Bohlmann et al. (2019) and Hirst (1952). During our campaign 18 different pollen types have been identified with the Hirst-type air sampler. The most abundant pollen types are birch (Betula), pine (Pinus) and spruce (Picea), which make up 93% of the total annual pollen count in 2019. Birch pollen grains are almost spherical, around 20-30 μm in diameter and have three pores on their surface. The surface pattern is rather smooth (Blackmore et al., 2003). Pine and spruce pollen belong to the Pinaceae family. Their pollen grains are large and non-spherical. They possess two air bladders which assist those pollen grains to be dispersed by wind despite their large size. The diameter of pine pollen grains on their longest axis is around 65-80 μm, while spruce pollen is considered very large for anemophilous pollen, with a diameter around 90-110 μm on the longest axis (Nilsson et al., 1977). The surface of the pollen corpus of spruce pollen is wrinkled and wavy (Grímsson and Zetter, 2011; Shen et al., 2020). Figure 1 shows examples of micrographs during our measurement period. Examples of birch and spruce pollen grains are marked with green and red arrows, respectively. The different size dimension of birch and spruce pollen is clearly visible."*

**4. Do you mainly observe whole pollen grains or fragments? A spruce pollen with 100 μm in size is large compared to all wavelengths used in your study and so no spectral behavior is expected. The strong difference in PLDR between 355 and 532 nm could not be explained with such large pollen grains. Your optical particle sizer (OPS) counts only up to 10 μm, which is even too small for birch pollen (20 – 30 μm in diameter). Do you have any indication about the strength of the fragmentation process? Is it related to temperature, RH, wind speed or age of the pollen grain?**

We cannot really say how much fragments we observe (please see also the reply to comment 6). With the Hirst sampler we only count the whole pollen grains. Pollen fragments were rarely seen, but the amount of fragments has not been recorded which would allow a quantitative analysis of the influence of fragments. The OPS limit is too small for birch pollen, but fragment could be detected if they fall within the range.

Only little research has been done on fragments of specific pollen types. It is likely that pollen fragments affect the detected depolarization and could explain the difference in the PLDR between 355 and 532

nm, but with the available data we cannot quantify the impact of pollen fragments. In future studies the impact of pollen fragments needs to be addressed and quantified by additionally collecting and counting pollen fragments.

The impact and uncertainties related to pollen fragments are now clarified in section 3.3 and the need for further studies of pollen fragments is mentioned in same section as well as the conclusion. Following paragraph was added in section 3.3:

*"Our results indicate that the influence of pollen fragments is uncertain as the amount of pollen fragments is not recorded by the Hirst-type pollen sampler and in-situ, meteorological and Halo turbulence measurements do not suggest apparent reasons for a pronounced fragmentation of pollen grains in the atmosphere. However, pollen fragments could cause depolarization in the lidar signal and also partly explain the wavelength dependence of the PDR. Therefore, the impact of pollen fragments needs to be explored in future studies."*

**5. Do the depolarization ratios reported by Cao et al. (2010) correspond to whole pollen grains or fragments?**

In the laboratory experiment by Cao et al. (2010), pollen is inserted in the aerosol chamber as dry powder and the pollen sample consist of whole pollen grains.

**6. Does the microscopic analysis of the Hirst sampler count only whole pollen grains or does it include fragments as well? Maybe the large spruce pollen grains are more likely fragmented. This could explain their high PLDR as you discuss in lines 330-333. Please extend the discussion and collect further information on fragmented pollen grains. Maybe on the tape of the Hirst sampler?**

The Hirst-type sampler efficiently traps particles down to 3.7 µm (Muilenberg, 2003). Pollen fragments which are smaller would not be collected by the device. The pollen tapes are manually analysed and counted by our colleagues at the University Turku. The tape has not been analysed for pollen fragments as the main purpose of the pollen collection is the determination of the present pollen type and concentration. This determination cannot be done from the pollen fragments; therefore pollen fragments are by default not counted. However, this could be one objective for future studies. This is now more clearly stated in section 2.3.

**7. Please improve your discussion of the spectral dependence of the PLDR and motivate, why it is important to measure the depolarization ratio at several wavelengths. The spectral dependence of the PLDR was studied previously for different aerosol types. A similar behavior with a maximum at 532 nm and a decrease towards the ultraviolet and the near infrared was observed for mineral dust (Burton et al., 2015, Haarig et al., 2017). The optical properties of mineral dust are dominated by the large particles (>1 µm) as well. Whereas small smoke particles in the stratosphere (< 1µm) show a completely different behavior with a strong decrease of the PLDR with increasing wavelength (Haarig et al., 2018, Hu et al., 2019).**

The discussion of the spectral dependence has been improved and moved to a separate Section (3.5). A comparison with the spectral dependence of other aerosol types was added as well.

The following new section was added:

*"3.5 Spectral dependence of the linear depolarization ratio*

*Figure 9 shows the daily mean linear particle depolarization ratios detected with PollyXT (at 355 and 532 nm) and the Halo Doppler lidar (at 1565 nm). The PDR at 355 nm is around 0.1 on all four days. From special interest is the strong spectral dependency of the PDR observed on 19 May. A higher depolarization ratio was observed at 532 than at 1565 nm. On the preceding days, however, the depolarization ratios at 532 and 1565 nm are only slightly higher than at 355 nm and almost equal. As shown in Table 1, the number of observed spruce pollen was increasing each day and contributed to almost 22 % of the total number concentration on 19 May. Longer wavelengths are more sensitive to larger particles and less influenced by Rayleigh scattering (Ansmann et al., 2009). The strong spectral dependency on 19 May can therefore be explained by the high amount of non-spherical spruce pollen on that day. Another explanation for the observed wavelength dependence could be the surface structure of pollen. Patterns on the outer wall of the pollen, the exine, can vary enormously in their appearance, but within the same species patterns are usually similar (Wang and Dobritsa, 2018). The non-smooth pattern on spruce pollen grains could additionally cause a depolarization of the backscattered light.*

*A spectral dependence of the linear depolarization ratio of pollen has already been reported by Cao et al. (2010) who measured the depolarization ratio for different pollen types, such as birch and pine, in controlled laboratory experiments. For birch pollen Cao et al. (2010) reported a PDR of 0.08 ± 0.01, 0.33 ± 0.00 and 0.30 ± 0.01 at 355, 532 and 1570 nm, respectively. No measurements have been conducted for spruce pollen. However, the authors state values for pine which belong to the same family as spruce and have smaller but in shape comparable pollen grains. According to Cao et al. (2010), depolarization ratios for pine pollen are 0.20 ± 0.01, 0.41 ± 0.01 and 0.28 ± 0.01 at 355, 532 and 1570 nm, respectively. The laboratory-derived depolarization ratios by Cao et al. (2010) are shown for comparison in Fig. 9. Empty markers represent the lidar depolarization ratios for paper birch (Betula papyrifera) and Virginia pine (Pinus virginiana). Especially, the spectral dependence for pine pollen is very similar to our observations on the spruce dominated day, having the peak at 532 nm. It should be kept in mind that the laboratory measurements were performed in an aerosol chamber and that they represent the depolarization ratio of the pure and dry pollen. Furthermore, pollen from different species were used compared to what we detected at our measurement site. Different values were therefore expected. However, the incidence that we detected similar wavelength dependence in ambient conditions as in the laboratory measurements, suggests that the wavelength dependence with maxima at 532 nm might be characteristic for pollen and especially for large and non-spherical conifer pollen types.*

*The spectral dependence of the PDR has previously been reported for different aerosol types. Haarig et al. (2018) and Hu et al. (2019) observed a completely different spectral dependence of the PDR for stratospheric smoke. The PDR is strongly decreasing with increasing wavelength caused by the dominance of small particles (< 0.5 µm). A similar spectral dependence of the PDR with the peak PDR at 532 nm and decreasing towards 355 and 1064 nm, however, was reported by Haarig et al. (2017) and Burton et al. (2015) for transported Saharan dust. As for pollen, the optical properties of dust are dominated by large particles (> 1µm) as well. However, depolarization ratios at 355 nm are clearly smaller for pollen than for dust, while the PDR at longer wavelengths is higher, which allows the distinction of pollen and aged dust."*

Furthermore, a motivation why the detection of multiple wavelengths is important was added in the introduction from lines 61-63 as:

*"The measurement of the depolarization ratio at multiple wavelengths allows us to investigate its wavelength dependence. This could enable the distinction of pollen from other depolarizing aerosols, as the depolarization ratio at single wavelengths can be similar for different aerosol types."*

**8. The depolarization ratio at 1565 nm is consequently not called particle depolarization ratio (PDR), but just depolarization ratio (DR). This might confuse some readers. In fact, you measure the volume depolarization ratio at 1565 nm. However, the molecular influence is negligible at such long wavelengths and the volume depolarization ratio is equal to the particle depolarization ratio. Can you calculate once the molecular influence at 1565 nm? Are gas absorption lines expect at this wavelength which could interfere with your results? With this low number you can argue, that you measure the PDR.**

The molecular depolarization cannot be observed with Halo as the pulse energy is too low and the wavelength too long to measure molecular scattering. There are no gas absorption lines at 1565 nm. The transmittance at this wavelength is 9.99999836E-1 according to HITRAN ([https://hitran.iao.ru/](https://hitran.iao.ru/)) using gas mixture (USA model, high latitude, summer, H=0).

A more detailed explanation about the consideration of the Halo depolarization ratio as particle depolarization ratio is given in (Vakkari et al., 2020). To avoid confusion we changed the naming of the Halo depolarization ratio to particle depolarization ratio throughout the manuscript in accordance with Vakkari et al. (2020).

**9. You are arguing that the pollen grains are large compared to the wavelength (355 and 532 nm), so the BAE = 0. But how do you explain the large difference in the PLDR between 355 and 532 nm?**

The wavelength dependence and the difference of the PDR at 355 and 532 nm could have different reasons which need more investigation. One explanation could be pollen fragments which cause the depolarization at 532 but not at the smaller wavelength. However, we were not able to count and quantify the impact of pollen fragments (see question 4 and 6). Another possible explanation could be the surface structure of pollen, which cause the wavelength dependence of the PDR but does not affect the backscattering and therefore the BÅE. Furthermore, the shorter wavelength could be more sensitive to small, spherical background aerosol.

Attempts to explain the wavelength dependence are now included in the case study discussion (Sec.3.1) and in section 3.3. Furthermore, we state that more studies are needed to identify the reason of the wavelength dependence with certainty.

**Minor comments**

**10. Ice nuclei (e.g., L.43) should be called "ice nucleating particles" (INP) to follow the convention of Vali et al., 2015. Please consider this throughout the manuscript.**

Thank you for this comment. The term "ice nuclei" is now changed to ice nucleating particle throughout the manuscript.

**11. L.39/40 "Airborne measurements revealed the presence of pollen up to 4 km above ground with a considerable amount of pollen still observed at 2 km" Where did they measure? Please include the region at this point.**

This statement refers to two examples of airborne pollen observations. Mandrioli et al. (1984) measured in Miles City, Montana (USA) during the Cooperative Convective Precipitation Experiment (CCOPE) using a light aircraft and detected pollen up to 4 km. Rempe (1937) measured in Göttingen and Kassel-Waldau (Germany) using a small aircraft. Because we consider one example of vertical (airborne)

measurements in this context as sufficient to show the vertical presence of pollen in the air and the article by Mandrioli et al. (1984) is more recent and published in English, we omit the example by Rempe (1937).

The region of the CCOPE experiment was added in the manuscript from lines 40-42 as:

"*Pollen can be lifted up to several kilometers by turbulent mixing within the boundary layer. Airborne measurements such as during the Cooperative Convective Precipitation Experiment (CCOPE) near Miles City, Montana (USA) revealed the presence of pollen up to 4 km above ground with a considerable amount of pollen still observed at 2 km (Mandrioli et al., 1984).*"

**12. L.45 The phrase "increase the cloud albedo and cloud lifetime and reduce precipitation" goes beyond the scope of your article and would need further references.**

We agree and removed this sentence.

**13. L.88 Did you merge the signals or the products? Please be more specific.**

We merged the products. It is now mentioned in the manuscript from lines 93-95 as:

"*Optical products retrieved from far- and near-field channels were merged to obtain the best possible profiles throughout the whole troposphere, enabling us to determine reliable profiles close to the ground.*"

**14. L.116 "was processed similar to Vakkari et al. (2020)" Where is the difference to Vakkari or is it just the different bleed-through? When did you perform the calibration at liquid cloud base?**

Only the bleed-through is different compared to the case studies presented in Vakkari et al. (2020). To clarify this, we changed the expression in line 116 to "was processed according to Vakkari et al. (2020)". Halo Doppler lidar bleed-through was estimated from liquid cloud base observed at 1200-1400 m a.g.l. on 14 June 0:30-3:30 UTC. This clarification was added in line 130-131:

"*The bleed-through in the Halo internal polarizer was estimated from liquid cloud base observed at 1200-1400 m a.g.l. on 14 June 0:30-3:30 UTC as 0.013 ± 0.006.*"

**15. L.148 The VDR in the UV is strongly influenced by the molecular scattering and therefore the effect of the particles is smaller compared to larger wavelengths.**

This explanation of the difference has been added at this location.

**16. L.155 Find a better title for the subsection. Something like "Case study with highly depolarizing aerosol".**

The title of the subsection was changed accordingly.

**17. Section 2.1, 3.1 and 3.2 needs some more structure, e.g., Section 2.1 starts and ends with depolarization. Better separate the discussion about the negative BAE from the spectral**

**dependence of PLDR in Section 3.1. In Section 3.2 move the pollen description to a separate (sub)section at an earlier position in the paper.**

Section 2.1:

In the beginning of this section the instrumental specifications of Polly$^{XT}$ are described, then the processing of the lidar data and the optical properties for aerosol characterization are shortly introduced. We consider the structure of this section logical and prefer to keep it like this.

Section 3.1:

This section is now separated from the spectral dependence, which is now in a new subsection (3.5).

Section 3.2:

The description of the pollen types was moved to a separate subsection (2.3) in Section 2.

**18. Which trees are surrounding your measurement site? Do they contribute to the pollen load?**

The measurement site is mainly surrounded by birch, spruce, and pine trees. It can be expected that the local trees contribute to the observed pollen load. Additional information about the surrounding trees is added in the manuscript in lines 70-71, as:

*"Vehmasmäki is surrounded by broadleaved and coniferous forest. Dominant tree species include Silver birch (Betula pendula), Norway spruce (Picea abies) and Scots pine (Pinus sylvestris)."*

**19. L.288 "It must be noted, that the Ångström exponent also depends on the background aerosols and its use to characterize pollen needs to be considered carefully". It is mentioned very late in the manuscript. At the beginning you should make clear, that BAE and PLDR are always a mixture of pollen and background aerosol. Already there, you can mention that you will later use the method of Shang et al. (2020) to separate the background and the pollen influence.**

It is now mentioned already when first presenting BÅE and PDR results that values always represent a mixture of pollen and background aerosol. Also reference to the later use of the method by Shang et al. (2020) is given.

We have modified lines 208-216 as:

*"In our observations, the depolarization ratios at 355 and 532 nm show a decline with height, while the Halo PDR at 1565 nm is almost constant throughout the whole boundary layer. Values of PDR, as well as BÅE, always represent a mixture of pollen and background aerosol. We assume that the concentration of pollen decreases with height, i.e. increasing distance to the pollen source, and the influence of background aerosol increases. The longer Halo wavelength is more sensitive to larger particles (pollen) and less sensitive to smaller particles (background aerosol) and thus the depolarization ratio does not decrease with height in contrast to the PDR at 355 and 532 nm. The different sensitivity to background aerosol could also explain the difference of the PDR at 355 and 532 nm. Later in this study, Sec. 3.6, we will use a method described by Shang et al. (2020) to separate the influence of background and the pollen on measured optical properties."*

**20. Fig. 3(e) Consider adjusting the x-axis to a maximum value of 2 to be comparable to the other backscatter coefficient profiles and enlarge the profiles for 16 – 18 May.**

Changing the scale would cut off the lowest part of the profile from 19 May. We therefore decide to only adjust the scale to a maximum value of 4.5 which slightly improves the visibility of the other profiles without cutting any profile.

**21. Fig. 4 Add the year to the dates in the caption.**

The year was added.

**22. Fig. 10 + 11 Please add the threshold (-0.4) to the figures.**

A vertical line to represent the threshold was added in Figure 10 and is now mentioned in the caption of Fig. 10 and 11.

**23. The text should be improved in terms of spelling and unusual expressions, e.g.,**

> **L.242 relationship, no plural.**

> **L.293 "Estimation of the pollen depolarization ratio"**

> **L.296 "of total particles" – find a better expression.**

> **L.302 "are related to PDR larger than"**

> **L.329 has –> have**

> **L.350 affect the detected depolarization ratio as well.**

All the suggestions have been applied to the manuscript.

**24. Avoid long sentences, better split them, e.g., L.85-88; 93-95 (using : : : using)**

These sentences were split into multiple sentences.

*Correspondence to*: Stephanie Bohlmann (stephanie.bohlmann@fmi.fi)

**Abstract.**

[revised manuscript text omitted]

50    importance to study the pollen distribution and transport mechanism in the atmosphere. To further improve pollen transport and forecast models, validation measurements are needed. Previously, we have shown that in the absence of other non-spherical particles, light detection and ranging (lidar) measurements and especially the particle depolarization ratio can be used to track pollen grains in the atmosphere (Bohlmann et al., 2019). We also estimated the depolarization ratio at 532 nm for pure birch (*Betula*) and pine pollen under certain assumptions as $0.24 \pm 0.01$ and $0.36 \pm 0.01$, respectively (Shang et al., 2020). However,

55    due to the measurement setup in our earlier studies, the lowest layers of the atmosphere (below 800 m), which naturally contain most of the airborne pollen, could not be investigated. In this study, we extend our dataset by operating an upgraded version of our multiwavelength Raman lidar Polly[XT] with four near-range channels at the same measurement site in Kuopio (Vehmasmäki), Finland. The improved lidar setup allowed measurements down to around 350 m above ground level. Furthermore, the new system measures the depolarization ratio also at 355 nm, while depolarization measurements were

60    limited to 532 nm in the previous campaign. To further improve the investigation of pollen depolarization properties, we assessed the depolarization measurements of a co-located Halo Doppler lidar, which provides information at 1565 nm (Vakkari et al., 2020). The measurement of the depolarization ratio at multiple wavelengths allows us to investigate its wavelength dependence. This could enable the distinction of pollen from other depolarizing aerosols, as the depolarization ratio at single wavelengths can be similar for different aerosol types.

65    Instrumentation and methods are presented in Section 2. The observation period is introduced in Section 3. Therein we present
      a case study and comparison of different pollen conditions during the period under study. Supporting measurement data, a
      statistical analysis, the discussion of the wavelength dependence and an estimation of pollen depolarization values are as well
      given in same section.

**2 Instrumentation and methods**

[revised manuscript text omitted]

In our observations,  the depolarization ratios at 355 and 532 nm show a decline with height, while the Halo PDR at 1565

215 nm is almost constant throughout the whole boundary layer. Values of PDR, as well as BÅE, always represent a mixture of pollen and background aerosol. We assume that the concentration of pollen decreases with height, i.e. increasing distance to the pollen source, and the influence of background aerosol increases. The longer Halo wavelength is more sensitive to larger particles (pollen) and less sensitive to smaller particles (background aerosol) and thus the depolarization ratio does not decrease with height in contrast to the PDR at 355 and 532 nm. The different sensitivity to background aerosol could also explain the

220 difference of the PDR at 355 and 532 nm. Later in this study, Sec. 3.6, we will use a method described by Shang et al. (2020) to separate the influence of background and the pollen on measured optical properties.

[revised manuscript text omitted]

**3.5 Spectral dependence of the linear depolarization ratio**

Figure 9 shows the daily mean linear particle depolarization ratios detected with Polly[XT] (at 355 and 532 nm) and the Halo

350 Doppler lidar (at 1565 nm). The PDR at 355 nm is around 0.1 on all four days. From special interest is the strong spectral dependency of the PDR observed on 19 May. A higher depolarization ratio was observed at 532 than at 1565 nm. On the preceding days, however, the depolarization ratios at 532 and 1565 nm are only slightly higher than at 355 nm and almost

equal. As shown in Table 1, the number of observed spruce pollen was increasing each day and contributed to almost 22 % of the total number concentration on 19 May. Longer wavelengths are more sensitive to larger particles and less influenced by

355  Rayleigh scattering (Ansmann et al., 2009). The strong spectral dependency on 19 May can therefore be explained by the high amount of non-spherical spruce pollen on that day. Another explanation for the observed wavelength dependence could be the surface structure of pollen. Patterns on the outer wall of the pollen, the exine, can vary enormously in their appearance, but within the same species patterns are usually similar (Wang and Dobritsa, 2018). The non-smooth pattern on spruce pollen grains could additionally cause a depolarization of the backscattered light.

360  A spectral dependence of the linear depolarization ratio of pollen has already been reported by Cao et al. (2010) who measured the depolarization ratio for different pollen types, such as birch and pine, in controlled laboratory experiments. For birch pollen Cao et al. (2010) reported a PDR of $0.08 \pm 0.01$, $0.33 \pm 0.00$ and $0.30 \pm 0.01$ at 355, 532 and 1570 nm, respectively. No measurements have been conducted for spruce pollen. However, the authors state values for pine which belong to the same family as spruce and have smaller but in shape comparable pollen grains. According to Cao et al. (2010), depolarization ratios

365  for pine pollen are $0.20 \pm 0.01$, $0.41 \pm 0.01$ and $0.28 \pm 0.01$ at 355, 532 and 1570 nm, respectively. The laboratory-derived depolarization ratios by Cao et al. (2010) are shown for comparison in Fig. 9. Empty markers represent the lidar depolarization ratios for paper birch *(Betula papyrifera)* and Virginia pine (*Pinus virginiana*). Especially, the spectral dependence for pine pollen is very similar to our observations on the spruce dominated day, having the peak at 532 nm. It should be kept in mind that the laboratory measurements were performed in an aerosol chamber and that they represent the depolarization ratio of the

370  pure and dry pollen. Furthermore, pollen from different species were used compared to what we detected at our measurement site. Different values were therefore expected. However, the incidence that we detected similar wavelength dependence in ambient conditions as in the laboratory measurements, suggests that the wavelength dependence with maxima at 532 nm might be characteristic for pollen and especially for large and non-spherical conifer pollen types.

The spectral dependence of the PDR has previously been reported for different aerosol types. Haarig et al. (2018) and Hu et

375  al. (2019) observed a completely different spectral dependence of the PDR for stratospheric smoke. The PDR is strongly decreasing with increasing wavelength caused by the dominance of small particles ($< 0.5$ µm). A similar spectral dependence of the PDR with the peak PDR at 532 nm and decreasing towards 355 and 1064 nm, however, was reported by Haarig et al. (2017) and Burton et al. (2015) for transported Saharan dust. As for pollen, the optical properties of dust are dominated by large particles ($> 1$µm) as well. However, depolarization ratios at 355 nm are clearly smaller for pollen than for dust, while the

380  PDR at longer wavelengths is higher, which allows the distinction of pollen and aged dust.

**3.65 Estimation of the pollen depolarization ratio values**

To estimate the depolarization ratio values for pure pollen, a newly developed method by Shang et al. (2020) was applied to the lidar measurements. Mathematically, there is a power law relationship between the total backscatter-related Ångström

385  exponent of total particles and the pollen backscatter contribution (the ratio of pollen backscatter coefficient and the total

particle backscatter coefficient). The depolarization ratio at 532 nm of pure pollen was investigated using the BÅE at 355/532 nm. Our results on this intense pollen event show two groups of data depending on the spruce pollen share: i) a quite good power law relationship is found for cases with a low share of spruce pollen; ii) the BÅE at 355/532 nm is almost constant with increasing PDR for cases with higher share of spruce pollen. These two groups can also be identified as the two modes in the

390    frequency distribution of BÅE values at 355/532 nm presented in Fig. 10. The first mode of values around -0.9, denoted as "extreme cases", are related to PDR  larger than 0.25, mostly for the data on the last day (high spruce share). For these cases the BÅE is almost constant throughout the boundary layer whereas the depolarization ratio is decreasing with height. This reveals the limitation of Ångström exponent at lidar wavelengths of 355 and 532 nm for the detection of big pollen particles since the BÅE normally decreases with increasing share of pollen particles, i.e. increasing PDR.

395    A threshold of -0.4 was applied to remove those "extreme cases" before applying the algorithm of Shang et al. (2020). A depolarization ratio of 0.03 at both 355 and 532 nm was used for non-pollen particles, which is a mean value for pollen-free periods at our measurement site. After the iteration, we found the relationship between the pollen BÅE at 355/532 nm and the pollen depolarization ratio at 532 nm presented in Fig. 11. This relationship is representing the mixture of birch and spruce pollen, with low spruce pollen share after the selection of the threshold. Our previous studies (Bohlmann et al., 2019; Shang

400    et al., 2020) indicate that pollen are medium to high absorbing particles with lidar ratio values from 55 to 70 sr at both 355 and 532 nm for birch and spruce pollen, without significant wavelength-dependence. The pollen grains are quite big, and thus can be assumed to be wavelength independent on the backscatter at wavelengths of 355 and 532 nm. Under the assumption that the backscatter-related Ångström exponent at 355/532 nm should be zero for pollen, the pollen depolarization ratio was found to be 0.44 , for the mixture of birch and spruce pollen observed during this intensive period of high pollen

405    concentrations. The spruce pollen share to the total pollen number concentration of the birch-spruce pollen mixture ranges between 0 and 35% for the estimated pollen depolarization value. The depolarization at 355 nm was calculated using the pollen backscatter contribution at 355 nm and the BÅE at 355/532 nm. Assuming a BÅE of zero, a pollen depolarization ratio of 0.16 at 355 nm was found for the birch and spruce mixture. In order to retrieve the pollen depolarization ratio of the "extreme cases", we used the BÅE at 532/1064 nm as longer lidar wavelengths might be beneficial for cases with a high share of very

410    large particles. However, our dataset of corresponding cases is too limited to lead to a meaningful conclusion. More cases with high spruce concentration are needed.

**Conclusion**

During the pollen campaign in 2019 at the EARLINET station in Vehmasmäki (Kuopio), Finland, we were able to expand the dataset of birch and spruce pollen observations with improved lidar-based measurements down to around 350 m. In previous

415    studies, measurements were restricted by a high overlap height which limited usable profiles to data above 800 m. The majority of strongly depolarizing spruce pollen, however, is assumed to be close to the pollen source near ground level. Thus, when conducting pollen measurements with lidar, the lower limit of usable lidar data should be as low as possible. In our study, we

utilized depolarization measurements of a co-located Halo Doppler lidar to add a third depolarization wavelength. Very high depolarization ratios have been detected during a 4-day period of birch and spruce pollen with an increasing amount of spruce
420 pollen towards the end. In-situ aerosol measurements and meteorological data has have been considered to characterize the overall aerosol load in the air and the air masses at the site. Ruptured pollen could affect the observed depolarization ratio as the smaller fragments of ruptured pollen are highly non-spherical. We detected a slightly higher number of particles in the diameter range from 3-10 µm on the day with high depolarization, which could be partly caused by pollen fragments. However, more research on the impact of pollen fragments is needed and the amount of pollen fragments should be recorded in future
425 studies. Also the possible impact of the pollen surface structure on the measured depolarization ratios needs to be investigated, as the rough surface of certain pollen types can cause higher depolarization ratios.

The investigation of lidar-retrieved optical properties in detected pollen layers revealed a wavelength dependence of the depolarization ratio especially in the presence of spruce pollen. On a day with a spruce pollen share of the total pollen number concentration of about 22%, high PDR values of $0.46 \pm 0.26$ and $0.30 \pm 0.09$ were detected at 532 and 1565 nm, respectively,
430 in pollen layers down to around 350 m. The PDR at 355 nm could only be detected down to 900 m. In those layers, mean PDR values were $0.10 \pm 0.02$, $0.38 \pm 0.23$ and $0.29 \pm 0.10$ at 355, 532 and 1565 nm, respectively. The wavelength dependency could be explained with the higher sensitivity of the longer lidar wavelengths to big pollen particles ($\sim 100$ µm) and characteristic surface pattern of those pollen. This andwavelength dependence could be characteristic for large, non-spherical spruce pollen. Furthermore, negative backscatter-related Ångström exponents have been detected when the spruce pollen share was high.
435 Negative Ångström exponents have previously been detected for dust but could also be a characteristic feature of spruce pollen. A novel method introduced by Shang et al. (2020) was applied to the measurement data and pollen depolarization ratio values of 0.44 at 532 nm and 0.16 at 355 nm were determined for the birch-spruce pollen mixture. Furthermore, a limitation of the Ångström exponent at the lidar wavelengths 355/532 nm for the characterization of very large pollen particles like spruce was found.
440 To determine values of pure pollen, more cases with high concentrations of only one pollen type are necessary. We found characteristic values for atmospheric conditions in the presence of spruce pollen, but at our measurement site spruce pollen almost always occurs simultaneously with other pollen types. This complicates the investigation of pollen type characteristics as optical properties vary depending on different mixtures of backscattering aerosol. Furthermore, pollen fragments and the pollen surface structure need to be considered when investigating pollen properties with lidar because non-spherical fragments
445 and pollen type-specific surface structures possibly affect the detected depolarization ratio as well. Nevertheless, our study shows the spectral dependence of the depolarization ratio for pollen, especially spruce, in the atmosphere.

**Data availability**

Lidar data are available upon request from the authors and data "quicklooks" are available on the PollyNET website
450 (http://picasso.tropos.de/, last access: 17/12/2020).

**Author Contributions:**

SB analyzed the data and wrote the manuscript in close cooperation with XS. VV provided the Halo Doppler lidar data. AL provided and assured the quality of the in-situ measurements. SP analyzed the pollen samples. MK and EG initiated the
455    project. All authors contributed to the scientific discussion and the article preparation.

**Conflicts of Interest:**

The authors declare no conflict of interest.

**Acknowledgments**

460    We acknowledge the Biodiversity Unit of University of Turku for the analysis of the pollen samples.

**Financial support:**

This work was supported by the Academy of Finland (project no. 310312).

[Figure]

620

**Figure 1 Pollen micrographs from 18.05.2019 (a) and 19.05.2019 (b) at 9-11 UTC. Examples of birch and spruce pollen grains are marked with green and red arrows, respectively (source: Biodiversity Unit of the University of Turku, Sanna Pätsi).**

625

[Figure]

**Figure  Overview of period 16-20 May 2019; Pollen concentration detect by Hirst-type spore trap at 4 m height (a), Range-corrected signal at 1064 nm (b), Linear volume depolarization ratio at 355 (c) and 532 nm (d) by Polly^{XT} Raman polarization lidar, Halo Doppler lidar depolarization ratio at 1565 nm (e) and automatic layer detection (f).**

[Figure]

630

**Figure  3 Case study on 19 May 2019. Range-corrected lidar signal at 1064 nm (a) and volume depolarization ratio at 532 nm (b). Time averaged profiles of backscatter coefficients (c), backscatter-related Ångström exponents (d) and depolarization ratios (e) of the selected time period, 11:00-13:00 UTC (white lines) are given. Profiles of temperature and relative humidity from GDAS1 data at 12 UTC (f).**

[Figure]

635

**Figure  4 Case studies from each day between 9:00 – 11:00 UTC. Backscatter coefficient and particle depolarization ratio at 355 nm and 532 nm by Polly^XT (a-d), attenuated backscatter and particle depolarization ratio at 1565 nm by Halo (e-f). The bar chart shows the pollen concentration during the 2-h period each day, with green representing birch and red representing spruce pollen.**

640

[Figure]

**Figure 5** Overview of pollen concentration by Hirst-type spore trap (a), air temperature and relative humidity at 2 m mast height (b), wind speed, wind direction and wind gust at 26 m mast height including wind direction shown by color (c) and turbulent kinetic energy (TKE) dissipation rate (d) by Halo Doppler lidar.

[Figure]

**Figure 6** Time series of particle number concentration (a) and volume size distribution in the range 0.3-10 μm (b) by optical particle sizer (OPS) using 10 min time average.

[Figure]

**Figure 7 Particle depolarization ratio (PDR) as measured with PollyXT at 355 vs. the PDR at 532 nm (a) and Halo PDR <s>depolarization ratio</s> at 1565 nm vs. Polly PDR at 532 nm (b). Color represents the share of the spruce pollen concentration to the total pollen count, size denotes the total pollen concentration. Black markers present daily mean values and their standard deviation.**

655

[Figure]

**Figure 8 Backscatter-related Ångström exponent (BÅE) at 355/532 nm (a) and 532/1064 nm (b) vs. the particle depolarization ratio (PDR) at 532 nm. Color represents the share of the spruce pollen concentration to the total pollen count, size denotes the total pollen concentration. Black markers present daily mean values and their standard deviation.**

660

[Figure]

**Figure 9 Spectral dependence of the mean particle depolarization ratio for each day. Additionally, mean linear depolarization ratio of pine and birch pollen determined in laboratory measurements by Cao et al. (2010) are shown. Error bars represent the standard deviation.**

665

[Figure]

$$\eta = (355/532)^{-B\mathring{A}E(355/532)}$$

**Figure 10 Frequency distribution of backscatter-related Ångström exponent at 355/532 nm of the pollen layers detected during the studied period. The related η value, as used in Shang et al.** (2020)**, is given as second x-axis. The BÅE threshold of -0.4 to remove "extreme cases" is marked by the dashed vertical line.**

670

[Figure]

$$\eta = (355/532)^{-B\mathring{A}E(355/532)}$$

**Figure 11 Assumed pollen backscatter-related Ångström exponent at 355/532 nm against the related pollen depolarization ratio at 532 nm for the studied period after removing cases below the BÅE threshold of -0.4. The pollen depolarization ratio is the calculated depolarization ratio value for a pollen backscatter contribution of 1 at assumed pollen BÅE.**

675

**Table 1 Daily mean values of pollen concentrations, and lidar derived optical parameters of the pollen layers. The total pollen (spruce) pollen) concentrations are measured by the Hirst-type volumetric air sampler. Layer heights are determined using the gradient method. Daily mean values and standard deviation of lidar-derived optical parameters in the pollen layers using a layer bottom height of 900 m (a) and 350 m (b).**

| Date in 2019 (dd.mm) | 16.05. | 17.05. | 18.05. | 19.05. |
|---|---|---|---|---|
| daily mean total (spruce) pollen concentration [#/m$^3$] | 3653 (8) | 3157 (63) | 1727 (132) | 3553 (776) |
| Layer top height (km) | $1.47 \pm 0.42$ | $1.41 \pm 0.23$ | $1.01 \pm 0.08$ | $1.23 \pm 0.41$ |
| (a) | | | | |
| PDR 355 | $0.08 \pm 0.03$ | $0.09 \pm 0.03$ | $0.11 \pm 0.03$ | $0.10 \pm 0.02$ |
| PDR 532 | $0.10 \pm 0.03$ | $0.16 \pm 0.10$ | $0.15 \pm 0.07$ | $0.38 \pm 0.23$ |
| PDR 1565 | $0.07 \pm 0.06$ | $0.16 \pm 0.14$ | $0.16 \pm 0.12$ | $0.29 \pm 0.10$ |
| BÅE 355/532 | $1.19 \pm 0.41$ | $0.92 \pm 0.39$ | $1.22 \pm 0.80$ | $-0.18 \pm 0.96$ |
| BÅE 532/1064 | $1.20 \pm 0.29$ | $1.18 \pm 0.74$ | $1.52 \pm 0.49$ | $1.08 \pm 0.46$ |
| Backscatter coefficient 355 [Mm$^{-1}$ sr$^{-1}$] | $0.44 \pm 0.15$ | $0.38 \pm 0.09$ | $0.48 \pm 0.14$ | $0.99 \pm 0.47$ |
| Backscatter coefficient 532 [Mm$^{-1}$ sr$^{-1}$] | $0.27 \pm 0.08$ | $0.26 \pm 0.06$ | $0.33 \pm 0.17$ | $1.27 \pm 0.86$ |
| (b) | | | | |
| PDR 532 | $0.14 \pm 0.03$ | $0.20 \pm 0.11$ | $0.22 \pm 0.11$ | $0.46 \pm 0.26$ |
| PDR 1565 | $0.13 \pm 0.06$ | $0.19 \pm 0.10$ | $0.21 \pm 0.10$ | $0.30 \pm 0.09$ |
| BÅE 355/532 | $1.29 \pm 0.52$ | $0.92 \pm 0.36$ | $0.90 \pm 0.68$ | $-0.10 \pm 0.92$ |
| Backscatter coefficient 355 [Mm$^{-1}$ sr$^{-1}$] | $0.44 \pm 0.12$ | $0.40 \pm 0.06$ | $0.62 \pm 0.16$ | $1.01 \pm 0.44$ |
| Backscatter coefficient 532 [Mm$^{-1}$ sr$^{-1}$] | $0.26 \pm 0.07$ | $0.28 \pm 0.05$ | $0.47 \pm 0.23$ | $1.26 \pm 0.85$ |

680